# Maternal Obesity Does Not Exacerbate the Effects of LPS Injection on Pregnancy Outcomes in Mice

**DOI:** 10.3390/biology9090293

**Published:** 2020-09-16

**Authors:** Natasha Virginkar, Julian K. Christians

**Affiliations:** 1Department of Biological Sciences, Simon Fraser University, 8888 University Drive, Burnaby, BC V5A 1S6, Canada; natasha_virginkar@sfu.ca; 2Centre for Cell Biology, Development and Disease, Simon Fraser University, 8888 University Drive, Burnaby, BC V5A 1S6, Canada; 3British Columbia Children’s Hospital Research Institute, 938 West 28th Avenue, Vancouver, BC V5Z 4H4, Canada; 4Women’s Health Research Institute, 4500 Oak Street, Vancouver, BC V6H 3N1, Canada

**Keywords:** obesity, inflammation, pregnancy, lipopolysaccharide, fetal growth, spontaneous abortion

## Abstract

**Simple Summary:**

Obesity increases the risk of problems during pregnancy, potentially due to inappropriate activation of the immune system. We predicted that, because of this immune activation, obesity in mice would exacerbate the effects of lipopolysaccharide (LPS), a substance that mimics infection, impairs fetal growth and leads to pregnancy loss in animal models. Mice were fed a high- or low-fat diet for thirteen weeks prior to mating, and then received an LPS or control injection during pregnancy. Treatment with LPS induced pregnancy loss in some mice, as expected. However, LPS did not have more severe effects in females fed the high-fat diet, who were heavier. Our results therefore do not support the hypothesis that an otherwise healthy obese pregnancy can be driven to an adverse outcome by a low-level infection. Our study improves the understanding of why obese women are at greater risk of adverse pregnancy outcomes. In doing so, it will contribute to future studies that seek to determine how obese pregnancies at risk of adverse outcomes can be distinguished from healthy obese pregnancies.

**Abstract:**

Obesity increases the risk of a number of pregnancy complications, potentially due to chronic inflammation. We predicted that an obesogenic high-fat diet (HFD) in mice would create an inflammatory environment that would exacerbate the effects of lipopolysaccharide (LPS), an inflammatory insult, administered during pregnancy. Females were placed on a HFD or a low-fat diet (LFD) prior to mating, injected with 2 µg LPS or control on gestational day 7 and collected on day 14. Treatment with LPS increased the odds that a female thought to be pregnant at injection had no conceptuses at day 14 (*p* = 0.024), suggesting that injection with LPS was more likely to induce complete abortion. However, there was no effect of diet on the odds of having no conceptuses at day 14 and no interaction between diet and LPS injection. Diet and LPS injection had no effect on the number of viable fetuses in females still pregnant at day 14. For fetal weight, there was a significant interaction between diet and treatment (*p* = 0.017), whereby LPS reduced fetal weight in HFD females but not in LFD females. However, LPS treatment of HFD females reduced fetal weight to that observed in control-injected LFD females. Although LPS increased the odds of abortion, there was little evidence that a HFD exacerbated the effects of LPS.

## 1. Introduction

Obesity doubles to triples the risk of pregnancy complications such as miscarriage, stillbirth and preeclampsia [1,2,3,4]. However, the mechanisms underlying associations between obesity and pregnancy complications remain unknown [5], and most obese women have healthy pregnancies. It has been suggested that inflammation, rather than obesity per se, affects the risk of adverse outcome [5,6]. Specifically, it is hypothesized that increased adipose tissue results in elevated levels of pro-inflammatory cytokines in the blood [7,8,9], which influence the function of immune cells within the uterus [5], thereby impairing normal placental development and increasing the risk of adverse outcomes [10,11]. While maternal obesity alone does not necessarily lead to complications, the associated chronic, low-grade inflammation might increase susceptibility to adverse outcomes when combined with an additional inflammatory challenge.

We sought to test this hypothesis in an animal model. We recently reported that an obesogenic high-fat diet (HFD) fed to mice altered populations and activity of uterine natural killer cells, maternal immune cells that play a crucial role in placental development [12]. However, this diet did not affect fetal weight and was not otherwise associated with adverse outcomes [12,13], suggesting that it models healthy obese pregnancy. We hypothesized that the immunological changes caused by this HFD could exacerbate the inflammatory effects of infection. Lipopolysaccharide (LPS), a component of the outer membrane of Gram-negative bacteria, is frequently used to induce inflammation in mice, and injection around gestational day (GD) 7-9 of pregnancy (where the day after mating is day 0) increases the rate of fetal resorption [14,15,16,17,18,19,20] and reduces fetal weight [14,19]. Few studies have combined HFD and LPS during pregnancy [21,22], but these examined adult offspring phenotypes rather than pregnancy outcomes and did not begin the HFD prior to pregnancy to induce maternal obesity. We predicted that maternal obesity induced by a HFD would exacerbate the response to a low dose of LPS, leading to synergistic effects on fetal growth and survival.

## 2. Materials and Methods

### 2.1. Mice

All work was carried out in accordance with the guidelines of the Canadian Council on Animal Care and approved by the SFU University Animal Care Committee (protocol 1188). C57BL/6J mice were purchased from the Jackson Laboratory (stock # 664) and were group-housed in individually ventilated cages (50 air changes/hour; max. 5 mice per cage) on a 12:12 h light:dark cycle at constant temperature (21 ± 1 °C), 50% humidity, with water and food available ad libitum. These mice were bred to produce the animals used in this study.

### 2.2. Diet Manipulation

At wean at three weeks of age, female offspring were randomly assigned to either a high-fat, high-sucrose diet (HFD; 45% kcal fat, 35% carbohydrate (including 17% kcal sucrose), 20% kcal protein, 4.73 kcal/g, D12451, Research Diets, New Brunswick, NJ, USA) or a nutrient-matched low-fat, no-sucrose control diet (LFD; 10% kcal fat, 70% kcal carbohydrate (corn starch and maltodextrin), 20% kcal protein, 3.85 kcal/g, D12450K, Research Diets). A nutrient-matched control diet was used instead of chow since the latter may differ from a defined HFD with regard to protein source and fiber content [23,24]. Females were maintained on the experimental diets for 13 weeks prior to breeding and food consumption was measured weekly (females were group-housed and so consumption could be calculated for cages but not individual mice). This HFD and duration of feeding had previously been found to increase maternal weight and adiposity with no effect on fetal growth [12,13]. More females were assigned to the HFD (N = 49) than to the LFD (N = 28) to enable separate analysis of HFD mice that were resistant to diet-induced obesity (DIO-R) or susceptible to diet-induced obesity (DIO-S). As described by others [25,26,27], we distinguished between the latter two groups based on whether the weight at first mating was lower (DIO-R) or higher (DIO-S) than 3 standard deviations above the mean weight at first mating of LFD mice that became pregnant.

### 2.3. Lipopolysaccharide Injection

Females were housed with a male overnight and separated from the male the following morning (gestational day (GD) 0). Males were fed normal chow (LabDiet Rodent 5001; 13.5% of calories from fat, 58% from carbohydrates and 28.5% from protein) and females were exposed to the male’s diet, not their experimental diet, for the overnight mating. We assessed pregnancy using weight gain rather than the presence of a vaginal plug as we have found the latter less reliable. On GD6, females were weighed, and those with substantial weight gain (2 g for HFD and 1 g for LFD mice) were considered pregnant. Different weight-gain thresholds for HFD and LFD mice were based on our previous experience with timed matings of mice on these diets [12,13]. On GD7, pregnant females were weighed again to confirm pregnancy and, if sufficient weight had been gained, were administered a subcutaneous injection of either 2 μg LPS (*Salmonella enterica*; L6511, Sigma-Aldrich, St. Louis, MO, USA) in 0.85% saline or the same volume of saline.

We used a constant dose of LPS per mouse, rather than a dose per unit of body weight, because the HFD mice were, on average, heavier. If we used dosed by weight, more severe effects in HFD/LPS treated mice might be due to a higher dose of LPS rather than exacerbation of the effects of LPS by a HFD. Dosing by weight would also result in HFD mice receiving a higher dose per lean mass. Furthermore, females were pregnant when LPS was administered, and, as a result, females carrying more fetuses would be treated with a higher dose. We aimed to use an LPS dose that, in lean mice, would have only moderate effects on fetal growth and resorption, so that it would be possible to discern if these were exacerbated by the HFD. We therefore used slightly lower doses than previous studies (e.g., 2.5 µg/mouse intraperitoneal at GD7 [15]; 0.5 µg/g subcutaneous at GD7 [14]; 0.3 µg/g subcutaneous at GD8 [17]). We used subcutaneous injection rather than intraperitoneal in an attempt to induce a chronic rather than acute response [28].

Females assessed to be not pregnant based on weight gain at GD6 were re-mated repeatedly until pregnant while maintained on their experimental diet. On GD14, females were sacrificed by carbon dioxide asphyxiation under isoflurane anesthesia. Uteri were examined for signs of fetal resorptions and other abnormalities, and the number of fetuses in each litter was recorded. Litter size was defined as the number of viable-appearing fetuses, not including the number of resorbed fetuses. Fetal resorptions were identified as dark masses that were substantially smaller than fetuses from the same litter. Entire uteri were immediately placed in a 4% PFA (paraformaldehyde) solution, stored at 4 °C for three days and then dissected in 1% phosphate-buffered saline to individually weigh fetuses and placentas. At the time of sacrifice, blood was collected by cardiac puncture and serum concentrations of TNF-α were later measured using a commercially available ELISA (DY410-05 R&D Systems, Minneapolis, MN, USA) with a 1:2 dilution factor according to the manufacturer’s protocol.

### 2.4. Statistical Analyses

All analyses were performed in SAS (Version 9.4; data provided in Appendix A; SAS code provided in Appendix A). Differences between means were tested using general linear models (proc GLM), and the numbers of viable-appearing fetuses and resorptions were also analyzed using nonparametric approaches (proc NPAR1WAY). Where a significant interaction between diet and LPS treatment was observed, the effect of LPS was tested separately within each diet (ESTIMATE statement). Since the dam was the unit of replication, we analyzed average fetal and placental weight per dam. The presence or absence of pregnancy at GD14 was analyzed using logistic regression (proc LOGISTIC). Age at pregnancy showed a skewed distribution and so was log-transformed prior to analysis. Means are presented ± standard error, except where noted.

## 3. Results

Of the mice mated (HFD N = 49; LFD N = 28), seven became pregnant but were mistakenly not injected (HFD N = 3; LFD N = 4), four were not observed to become pregnant after repeated matings and were culled (HFD N = 2; LFD N = 2), and one HFD female (that had not been injected) was found dead. Over the first 13 weeks of the experimental diets, HFD females consumed more calories, more protein, more fat and less carbohydrate than LFD females (Table 1). At the time of the first mating, HFD females were heavier than LFD females (HFD: 25.5 ± 0.4 g; LFD: 22.8 ± 0.5 g; *p* < 0.0001).

A substantial number of females injected on GD7 were subsequently found to be not pregnant when dissected following sacrifice at GD14 (Table 2). Using logistic regression, we tested whether the odds that a female thought to be pregnant and injected at GD7 had no conceptuses at GD14 were influenced by treatment (LPS vs. saline), diet or the interaction between treatment and diet, controlling for weight gain between mating and GD6 (we recorded weights at GD6 and only confirmed weight gain at GD7 prior to injection). The odds of not being pregnant were significantly higher for females injected with LPS (adjusted OR = 3.3, 95% confidence interval: 1.1–10.1; *p* = 0.024; adjusted OR from model without diet by treatment interaction), suggesting that LPS was more likely to induce complete pregnancy loss. This analysis controlled for weight gain, where greater weight gain was associated with reduced odds of not being pregnant (*p* = 0.014). However, there was no effect of diet (*p* = 0.77) or interaction between diet and treatment (*p* = 0.28), i.e., no evidence that diet exacerbated the effects of LPS. In no case was a female with an apparent pregnancy loss mated again, i.e., all data are from first apparent pregnancies.

Among females with confirmed pregnancy at sacrifice at GD14, there was no effect of diet on the age at which they became pregnant, although there was a marginally significant decrease in the number of pairings that HFD females required to become pregnant (Table 3). In this group of animals, HFD females were heavier than LFD females at the time of mating resulting in pregnancy, but HFD females lost weight between their first mating and the mating when they became pregnant, whereas LFD females did not (Table 3).

Average fetal weight and placental weight were not affected by diet or treatment, controlling for female age (Table 4). However, the interaction between diet and treatment had a significant effect on fetal weight (*p* = 0.017), whereby LPS reduced fetal weight in HFD females (difference between LPS and saline: 56 ± 2 mg; *p* = 0.03) but not in LFD females (difference between LPS and saline: −51 ± 3 mg; *p* = 0.14; Table 4). There was no diet by treatment interaction for placental weight. Fetal weight decreased with increasing female age. There was no effect of diet, treatment or diet by treatment interaction on the number of viable-appearing fetuses or the number of resorptions (Table 4). The number of viable-appearing fetuses decreased, and the number of resorptions increased, with increasing female age. Serum TNF-α was not detectable in 11 out of 27 samples analyzed and showed no effect of diet or treatment (Appendix A).

Of the 20 HFD females where pregnancy was confirmed, 11 weighed less at first mating than the mean of LFD mice plus 3 SD (22.3 g + 3 × 1.3 g) and so were deemed to be resistant to diet-induced obesity. When separating the HFD group into females that were susceptible or resistant to diet-induced obesity, we obtained similar results (Table 5) as when all HFD females were grouped together (Table 4).

## 4. Discussion

It has been hypothesized that the increased risk of pregnancy complications in obesity is due to inflammation [5,6]. Obese pregnancies are associated with increased systemic and placental inflammation [29,30] and alterations in immune cell populations that affect placental development [31,32]. Potentially as a result, obesity is associated with abnormal uterine artery remodeling [33]. In addition to effects within the placenta, obesity may also increase the responsiveness of adipose tissue to endotoxins including LPS [29]. In mice, diet-induced obesity also affects immune cell populations [12,34], as well as placental development [35,36]. Outside of pregnancy, mice fed a HFD show an increased response to LPS [37].

These observations led us to predict that diet-induced obesity would exacerbate the effects of LPS administered during pregnancy. Treatment with LPS increased the odds that a female thought to be pregnant at injection had no conceptuses at day 14, suggesting that LPS injection was more likely to induce complete abortion than the control injection. However, this effect was not exacerbated by a HFD. LPS had an all-or-nothing effect on abortion, with no effect on the number of viable-appearing fetuses or fetal resorptions among females who remained pregnant. There was a significant interaction between diet and treatment, whereby LPS reduced fetal weight in HFD females but not in LFD females. While this might suggest synergy between LPS and HFD, LPS treatment of HFD females reduced fetal weight to that observed in control-injected LFD females, i.e., the combination of HFD and LPS did not result in growth restricted fetuses. These effects on fetal weight were similar regardless of whether females were resistant or susceptible to diet-induced obesity, indicating that they were the result of diet or other metabolic alterations rather than weight per se.

We did not observe effects of the HFD on serum TNF-α, but our prediction does not rest on an assumption of systemic inflammation. Rather, we have previously observed that this HFD alters the activity of uterine natural killer cells [12], and we predicted that this would render the pregnancy more susceptible to the inflammatory insult of LPS. Our previous studies have found that this HFD elevates circulating TNF-α and leptin levels [12,13], but it is possible that a HFD may not lead to systemic inflammation in a consistent manner.

## 5. Conclusions

We administered LPS at a dose sufficient to cause pregnancy loss in some mice but did not find that the effects of LPS were exacerbated by a HFD. Our results therefore do not support the hypothesis that an otherwise healthy obese pregnancy can be driven to an adverse outcome by an inflammatory insult. To our knowledge, this is the first experimental test of this hypothesis. However, since LPS from a single bacterial species models just one aspect of an inflammatory or infectious challenge, further studies combining diet-induced obesity with additional challenges are necessary.

## Figures and Tables

**Table 1 biology-09-00293-t001:** Food consumption of females fed a high-fat diet (HFD) or low-fat diet (LFD) for 13 weeks prior to mating. Females were group-housed and so consumption was calculated for cages, not individual mice.

	LFD (N = 6)	HFD (N = 10)	*p*-Value
Calories (kcal/mouse/day)	10.2 ± 0.4	11.6 ± 0.3	0.01
Protein (g/mouse/day)	0.51 ± 0.02	0.59 ± 0.02	0.005
Fat (g/mouse/day)	0.11 ± 0.02	0.59 ± 0.01	0.0001
Carbohydrate (g/mouse/day)	1.77 ± 0.04	1.01 ± 0.03	0.0001

**Table 2 biology-09-00293-t002:** Numbers of females fed a high-fat diet (HFD) or low-fat diet (LFD) for 13 weeks prior to mating, injected with lipopolysaccharide (LPS) or saline on gestational day 7 and found to be pregnant or not pregnant on gestational day 14.

Injection	Diet	Pregnant at GD14		Appeared Pregnant at GD7 But No Evidence of Pregnancy on GD14	
LPS	HFD	9	14	13	22
	LFD	5		9	
Saline	HFD	11	17	10	12
	LFD	6		2	

**Table 3 biology-09-00293-t003:** Characteristics of females fed a high-fat diet (HFD) or low-fat diet (LFD) for 13 weeks prior to mating and confirmed to be pregnant at GD14.

	LFD (N = 11)	HFD (N = 20)	*p*-Value
Age at which females became pregnant (days) ^1^	184 ± 1	167 ± 1	0.13
Number of times paired with male to achieve pregnancy ^2^	5.5 ± 0.6	3.4 ± 0.5	0.05
Weight at first mating (g)	22.3 ± 0.7	26.3 ± 0.5	0.0001
Weight at mating that resulted in pregnancy (g)	22.5 ± 0.5	24.7 ± 0.4	0.001
Weight change ^3^ (g)	−0.2 ± 0.6	1.5 ± 0.4	0.02

^1^ Age at which females became pregnant was log-transformed for analysis and back-transformed for presentation, which is why the standard error appears very small. ^2^ Number of times paired with male to achieve pregnancy was analyzed with nonparametric Wilcoxon and Kruskal–Wallis tests. ^3^ Weight change = weight at first mating–weight at mating that resulted in pregnancy.

**Table 4 biology-09-00293-t004:** Reproductive traits in females fed a high-fat diet (HFD) or low-fat diet (LFD) for 13 weeks prior to mating, injected with LPS or saline on gestational day 7 and found to be pregnant at gestational day 14.

	LFD		HFD		*p*-Value			
	LPS (N = 5)	Saline (N = 6)	LPS (N = 9)	Saline (N = 11)	Diet	Treatment	Diet * Treatment ^1^	Female Age
Fetal weight (mg)	252 ± 26	201 ± 22	203 ± 18	259 ± 16	0.83	0.90	0.017	0.016 ^2^
Placental weight (mg)	144 ± 8	143 ± 7	149 ± 6	147 ± 5	0.50	0.86	0.95	0.40
Viable-appearing fetuses	6.1 ± 0.9	6.2 ± 0.8	5.7 ± 0.6	6.2 ± 0.6	0.77 ^3^	0.67 ^3^	0.76	0.011 ^2^
Resorptions	1.2 ± 0.6	1.3 ± 0.6	2.1 ± 0.5	1.4 ± 0.4	0.39 ^3^	0.51 ^3^	0.46	0.008 ^2^

* the standard notation for a statistical interaction. ^1^ The statistical interaction between diet and treatment (LPS vs. saline). ^2^ For fetal weight and number of viable-appearing fetuses, the relationship with female age was negative, while for number of resorptions, it was positive. ^3^ Effects were not significant when testing diet or treatment separately with nonparametric Wilcoxon or Kruskal–Wallis tests.

**Table 5 biology-09-00293-t005:** Reproductive traits in females fed a high-fat diet (HFD) or low-fat diet (LFD) for 13 weeks prior to mating, injected with LPS or saline on gestational day 7 and found to be pregnant at gestational day 14. Females that weighed more or less at first mating than the mean of LFD mice plus 3 SD were considered susceptible or resistant to diet-induced obesity (DIO-S or DIO-R), respectively.

	LFD		DIO-R		DIO-S		*p*-Value			
	LPS (N = 5)	Saline (N = 6)	LPS (N = 4)	Saline (N = 7)	LPS (N = 5)	Saline (N = 4)	Diet	Treatment	Diet * Treatment ^1^	Female Age
Fetal weight (mg)	253 ± 26	201 ± 23	190 ± 29	259 ± 21	212 ± 25	257 ± 28	0.93	0.32	0.06	0.018 ^2^
Placental weight (mg)	145 ± 8	143 ± 7	139 ± 9	150 ± 6	157 ± 7	142 ± 8	0.73	0.82	0.27	0.28
Viable-appearing fetuses	6.1 ± 0.9	6.2 ± 0.8	5.8 ± 1.0	6.2 ± 0.8	5.6 ± 0.9	6.3 ± 1.0	0.97 ^3^	0.60	0.95	0.018 ^2^
Resorptions	1.2 ± 0.7	1.3 ± 0.6	2.6 ± 0.7	1.6 ± 0.5	1.8 ± 0.6	1.0 ± 0.7	0.41 ^3^	0.29	0.67	0.007 ^2^

* the standard notation for a statistical interaction. ^1^ The statistical interaction between diet and treatment (LPS vs. saline). ^2^ For fetal weight and number of viable-appearing fetuses, the relationship with female age was negative, while for number of resorptions, it was positive. ^3^ Effects were not significant when testing diet separately with nonparametric Kruskal–Wallis test.

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
