# Peer review of "Maternal Obesity Does Not Exacerbate the Effects of LPS Injection on Pregnancy Outcomes in Mice"

_biology, 2020, doi:10.3390/biology9090293_

Round 1
Reviewer 1 Report
Obesity is an increasing epidemic in developed countries and is said to increase the risk of pregnancy complications. The objective of this study was to determine if obesity exacerbated the adverse pregnancy outcomes associated with infection. The authors conclude that obesity had no effect on pregnancy outcomes after LPS treatment.
The manuscript is very well written and is relatively straightforward. There are a few gaps in the study that could increase impact and some results could be presented more clearly.
- How does the LFD compare to a normal mouse show diet in content? If the LFD is different from a normal mouse chow, how might this affect the outcomes?
- Did the authors measure adiposity in the animals to confirm that obesity in HFD dams compared to LFD?
- Can the authors measure LPS induced inflammatory cytokines (IL-1 beta) in the plasma to confirm the induction of inflammation by the LPS treatment? Likewise, can inflammatory cytokines or adipokines be measured in the LFD vs HFD animals to compare the "low grade" inflammation described to be present in obesity. It is important to demonstrate that the model is truly replicating the condition described.
- Table 1 is confusing. Please re-do the table to more clearly convey the results. As it is written, it seems that some of the animals did not get pregnancy. It was not clear that the "Not Pregnant" column was referring to the status of the animals on gestational day 14.
Author Response
Response to Reviewer 1
Point 1: Obesity is an increasing epidemic in developed countries and is said to increase the risk of pregnancy complications. The objective of this study was to determine if obesity exacerbated the adverse pregnancy outcomes associated with infection. The authors conclude that obesity had no effect on pregnancy outcomes after LPS treatment.
The manuscript is very well written and is relatively straightforward. There are a few gaps in the study that could increase impact and some results could be presented more clearly.
How does the LFD compare to a normal mouse show diet in content? If the LFD is different from a normal mouse chow, how might this affect the outcomes?
Response 1:
We thank the reviewer for their constructive comments.
Rodent chow varies somewhat between facilities, but the chow used at our facility (5001 from LabDiet) provides 13.5% of calories from fat, 58% from carbohydrates and 28.5% from protein, whereas the chow used with breeding females (Prolab RMH 3000), provides 14%, 60% and 26% from fat, carbohydrates and protein respectively. These diets are therefore much closer to our LFD (10%, 70%, 20%) than to our HFD (45%, 35%, 20%). Importantly, our LFD is nutrient matched to the HFD, such that only the fat and carbohydrate content differs. This is crucial, since chow will differ from a defined HFD in terms of protein source and fiber content in addition to fat and carbohydrate. Thus, using chow as the control diet results in a manipulation of protein source and fiber content in addition to fat and carbohydrate. The rationale for the use of a nutrient-matched control diet has been added at line 67.
Point 2: Did the authors measure adiposity in the animals to confirm that obesity in HFD dams compared to LFD?
Response 2:
Unfortunately the pandemic precluded dissection of females to weigh fat depots. However, we do not feel that these data are important for the interpretation of our data for a number of reasons. Firstly, human obesity is generally assessed using weight (i.e., BMI). Secondly, we have previously found that this diet increases adiposity (described at line 72). Thirdly, we observed similar effects whether analysing all HFD females together or analysing females that gained substantial weight (DIO-S) separately from those who were similar in weight to LFD females (DIO-R). Finally, we would have been able to measure adiposity only at the time of collection, and not at the time of LPS treatment; adiposity might be expected to change over 7 days of gestation.
Point 3: Can the authors measure LPS induced inflammatory cytokines (IL-1 beta) in the plasma to confirm the induction of inflammation by the LPS treatment? Likewise, can inflammatory cytokines or adipokines be measured in the LFD vs HFD animals to compare the "low grade" inflammation described to be present in obesity. It is important to demonstrate that the model is truly replicating the condition described.
Response 3:
The effects of LPS on circulating inflammatory cytokines is short-lived (< 1 day, see Fig. 5 in https://www.ncbi.nlm.nih.gov/pmc/articles/PMC7210116/ or Fig. 3 in https://www.ncbi.nlm.nih.gov/pmc/articles/PMC2186306/). Given that LPS was injected at GD7, we would not expect to see an effect of LPS at GD14 when we collected mice. We didn’t blood sample within a day after LPS injection because we did not want to disturb pregnancy further.
Although we did not expect to see an effect of LPS, we did measure serum TNF-alpha to assess whether diet affected inflammation. Using the same diet model, we previously found TNF-alpha levels to be higher at GD10 in HFD mice (Baltayeva et al. 2020, cited in manuscript). However, in the present study we sampled females at GD14 and found TNF-alpha was not detectable in 11 out of 27 samples analysed. Upon further investigation, we found that it is not unusual for TNF-alpha to be undetectable in mouse serum, e.g.,
https://www.ncbi.nlm.nih.gov/pmc/articles/PMC5845433/
https://onlinelibrary.wiley.com/doi/full/10.1038/oby.2009.445
https://www.ncbi.nlm.nih.gov/pmc/articles/PMC6351076/
We had not previously included these data in the manuscript because we felt they were inconclusive, but have added them to the revision (lines 109 and 169). Because we measured TNF-alpha with a 1:2 dilution factor, we used up most of the serum for most samples and are therefore not able to measure other inflammatory cytokines.
Although we did not find an effect of the HFD on TNF-alpha levels, we feel this negative result is useful given the widespread assumption that obesity/ high-fat diets are associated with inflammation.
Point 4: Table 1 is confusing. Please re-do the table to more clearly convey the results. As it is written, it seems that some of the animals did not get pregnancy. It was not clear that the "Not Pregnant" column was referring to the status of the animals on gestational day 14.
Response 4: The headings in Table 1 (Table 2 in revised manuscript) have been revised.
Reviewer 2 Report
This is an intesting paper presenting the results of inducing an inflammatory response in obesity mice on pregnancy outcomes. There are a number of queries that should be addressed before the paper is considered for publication.
- What was the degres of obesity. Althoigh the dams are fed a high fat high sugar diet there are not measures of obesity - TDMNR, fat content of dams. Especially as they are loosing weight between matings some indication of the level of pbesity needs to be provided at the time of mating. As the HFD dams are subsequently divided into susceptible and resistant to diet induced obesity some measure of obesity is essential - not just weight.
- Lines 63-74. Please give food intake as well as composition of the diet as the induction of an inflammatory state may have effects on food intake which in turn is likely to affect fetal and placental growth. In addition calculation of CHO, fat and protein intake would also add significantly yo the paper if the authors know the actual amount of food eaten.
- Line 76. Please give composition of the chow diet fed to the males. Were thje males on the HFD diet for the period of mating or were the dams transferred to the chow diet?
- Line 96. What was the average number of times the dams had to be mated to obtain a pregnancy? Did this vary with diet? Were feamles remated that initially gained weight and then lost the litter as this may bias the results as the dam would have experiences the endocrine changes of early pregnancy which could have affected the next? Second pregnancies are known to be more successful in reaching term than first.
- Line 115. What is meant be pregnany outcome?
- Was sex of the fetus considered in the statististical analyses as many other studies including the authors own have shown that the feto-placental response to environmental stresses is sex-linked. Were the sex ratios within litters changed with treatment?
- Table 2. The dams seemed to take along time to become pregnant even on the LFD. Mating started at 16 weeks but avaerage pregnancy attained at 24+ weeks. This suggests that the even the LFD was having adverse effects on reproductive function.
- Why were the fetuses smaller in the saline treated LFD dams than in their LPS treated counterparts? Some discussion of this is needed in the Discussion section and may relate to comment 1 and 2 and the degree of obesity and actual food intake in the different groups of dams. It suggests that the LFD is sub-optimal for fetal growth in its own right.
Author Response
Response to Reviewer 2
Point 1: This is an intesting paper presenting the results of inducing an inflammatory response in obesity mice on pregnancy outcomes. There are a number of queries that should be addressed before the paper is considered for publication.
What was the degres of obesity. Althoigh the dams are fed a high fat high sugar diet there are not measures of obesity - TDMNR, fat content of dams. Especially as they are loosing weight between matings some indication of the level of pbesity needs to be provided at the time of mating. As the HFD dams are subsequently divided into susceptible and resistant to diet induced obesity some measure of obesity is essential - not just weight.
Response 1:
We thank the reviewer for their constructive comments.
Unfortunately the pandemic precluded dissection of females to weigh fat depots. However, we do not feel that these data are important for the interpretation of our data for a number of reasons. Firstly, human obesity is generally assessed using weight (i.e., BMI). Secondly, we have previously found that this diet increases adiposity (described at line 72). Thirdly, we observed similar effects whether analysing all HFD females together or analysing females that gained substantial weight (DIO-S) separately from those who were similar in weight to LFD females (DIO-R). Finally, we would have been able to measure adiposity only at the time of collection, and not at the time of LPS treatment; adiposity might be expected to change over 7 days of gestation.
Point 2: Lines 63-74. Please give food intake as well as composition of the diet as the induction of an inflammatory state may have effects on food intake which in turn is likely to affect fetal and placental growth. In addition calculation of CHO, fat and protein intake would also add significantly yo the paper if the authors know the actual amount of food eaten.
Response 2:
These data have been added (described at lines 70, 126 and a new Table 1).
Point 3: Line 76. Please give composition of the chow diet fed to the males. Were thje males on the HFD diet for the period of mating or were the dams transferred to the chow diet?
Response 3:
The chow composition has been added (line 80). Females were transferred to the chow diet overnight (clarified on line 81).
Point 4: Line 96. What was the average number of times the dams had to be mated to obtain a pregnancy? Did this vary with diet?
Response 4:
These data have been added (described at lines 149 and Table 3). The difference between diets was marginally significant (P = 0.0507 with Wilcoxon test, P = 0.0483 with Kruskal-Wallis test).
Point 5: Were feamles remated that initially gained weight and then lost the litter as this may bias the results as the dam would have experiences the endocrine changes of early pregnancy which could have affected the next? Second pregnancies are known to be more successful in reaching term than first.
Response 5:
Females that gained and lost weight (presumably due to pregnancy loss) were collected at GD14. In no case was a female with an apparent pregnancy loss remated, i.e., all data are from first apparent pregnancies. This has been clarified (line 142).
Point 6: Line 115. What is meant be pregnany outcome?
Response 6:
This has been revised to “The presence or absence of pregnancy at GD14” (Line 119).
Point 7: Was sex of the fetus considered in the statististical analyses as many other studies including the authors own have shown that the feto-placental response to environmental stresses is sex-linked. Were the sex ratios within litters changed with treatment?
Response 7:
We initially intended to include fetal sex in the analyses. Unfortunately, the fixation of the uteri in 4% PFA impaired the quality of the DNA extracted from the fetuses (we determine fetal sex by PCR), and the pandemic precluded attempts to troubleshoot this issue.
Point 8: Table 2. The dams seemed to take along time to become pregnant even on the LFD. Mating started at 16 weeks but avaerage pregnancy attained at 24+ weeks. This suggests that the even the LFD was having adverse effects on reproductive function.
Response 8:
The length of time to become pregnant was due to our mating strategy, i.e., pairing a female with a male for one night only and waiting a week to assess if the female was pregnant. Females were not necessarily mated every week to (a) avoid mating large numbers which might result in having to collect an unfeasible number of females on a single day, and (b) limit the number of females with a single male at a time. Thus, the amount of time to become pregnant does not provide a measure of reproductive function.
Point 9: Why were the fetuses smaller in the saline treated LFD dams than in their LPS treated counterparts? Some discussion of this is needed in the Discussion section and may relate to comment 1 and 2 and the degree of obesity and actual food intake in the different groups of dams. It suggests that the LFD is sub-optimal for fetal growth in its own right.
Response 9:
In LFD females, the difference in fetal weight between LPS and saline treated females was not significant, and so we feel that it could be misleading to discuss it. Even if LPS did increase fetal weight in LFD dams (again, this was not significant), this would be a curious result but would not suggest that the LFD is sub-optimal for fetal growth, but rather that LPS somehow stimulated overgrowth. This particular LFD is widely used as a control diet (e.g., see Supplemental Table 2 in our review https://rbej.biomedcentral.com/articles/10.1186/s12958-019-0482-y and search for “12450”). While LFD females required more pairings to become pregnant, this result was only marginally significant and, if real, could be due to effects on behaviour, ovulation or uterine receptivity, i.e., these data are not evidence that LFD is sub-optimal for fetal growth.